# Recent Advances in Anti-Tuberculosis Drug Discovery Based on Hydrazide–Hydrazone and Thiadiazole Derivatives Targeting InhA

**DOI:** 10.3390/ph16040484

**Published:** 2023-03-23

**Authors:** Yoanna Teneva, Rumyana Simeonova, Violeta Valcheva, Violina T. Angelova

**Affiliations:** 1Faculty of Pharmacy, Medical University of Sofia, 1000 Sofia, Bulgaria; 2The Stephan Angeloff Institute of Microbiology, Bulgarian Academy of Sciences, 1113 Sofia, Bulgaria

**Keywords:** tuberculosis, InhA, hydrazide-hydrazones, thiadiazole-containing derivatives, *Micobacterium tuberculosis* resistance

## Abstract

Tuberculosis is an extremely serious problem of global public health. Its incidence is worsened by the presence of multidrug-resistant (MDR) strains of Mycobacterium tuberculosis. More serious forms of drug resistance have been observed in recent years. Therefore, the discovery and/or synthesis of new potent and less toxic anti-tubercular compounds is very critical, especially having in mind the consequences and the delays in treatment caused by the COVID-19 pandemic. Enoyl-acyl carrier protein reductase (InhA) is an important enzyme involved in the biosynthesis of mycolic acid, a major component of the *M. tuberculosis* cell wall. At the same time, it is a key enzyme in the development of drug resistance, making it an important target for the discovery of new antimycobacterial agents. Many different chemical scaffolds, including hydrazide hydrazones and thiadiazoles, have been evaluated for their InhA inhibitory activity. The aim of this review is to evaluate recently described hydrazide-hydrazone- and thiadiazole-containing derivatives that inhibit InhA activity, resulting in antimycobacterial effects. In addition, a brief review of the mechanisms of action of currently available anti-tuberculosis drugs is provided, including recently approved agents and molecules in clinical trials.

## 1. Introduction

Tuberculosis (TB) is a major communicable disease with serious adverse effect on both global health and the global economy. TB was the leading global cause of mortality due to infectious disease before the COVID-19 pandemic. The unprecedented pandemic negatively impacted the implementation of national and international TB control programs, particularly by the disruption of timely diagnostic and treatments, decreases in notification rates, insufficient follow-up with patients, and finally, increases in mortality. In order to counteract this situation, more efforts and resources must be allocated. According to the Global Tuberculosis Report 2022 [1], the COVID-19 pandemic has prevented people from accessing TB diagnosis and treatment and has slowed down the progress in the treatment of the disease. A diagnostic delay leads to a more severe disease, with increased risks of spreading and mortality. An estimated 10.6 million developed TB in 2021, compared to 10.1 million in 2020. Regarding mortality, 1.6 million people died from TB in 2021 (including 187,000 HIV-coinfected individuals), compared to 1.5 million in 2020 (including 214,000 HIV-coinfected individuals) [1].

A major clinical problem in treating TB is that the number of drugs available is limited. This problem becomes even more severe in the treatment of MDR/XDR-TB. This concerning situation is due, in part, to an almost 50-year-long drought in anti-TB drug development. Overall, recent developments have not been sufficient and have not yet addressed the main shortcomings of long, toxic treatments and the rise of drug resistance.

The WHO’s recommendations for people infected with drug-susceptible strains of TB have been set out in guidelines that were published in 2010 and updated in 2017 and 2022 [2]. The recommended six-month regimen includes four key first-line anti-TB drugs: isoniazid, rifampicin, ethambutol, and pyrazinamide. In addition, the 2022 guidelines considered new regimens: (i) the six-month regimen based on bedaquiline, pretomanid, and linezolid (BPaL) in combination with moxifloxacin (BPaLM), which was evaluated in the TB-PRACTECAL randomized clinical trial; (ii) the six-month regimens based on the BPaL combined with decreased exposure to linezolid (lower dosing or shorter duration), which was evaluated in the ZeNix study; and (iii) modified all-oral shorter regimens (6–9 months or 9–12 months) that included all three Group A drugs, which were evaluated in the NeXT trial or by using programmatic data from South Africa [3]. While the above regimens have been developed for treating drug-susceptible TB, the emergence and spread of drug-resistant strains is of global concern and presents challenges for TB control and treatment. *M. tuberculosis* strains can acquire drug resistance through various molecular mechanisms, including mutations in gene-encoding drug targets or drug-activating enzymes. Compensatory mutations and the activation of efflux pumps also contribute to the development of mycobacterial drug resistance [4].

Figure 1 presents the chemical scaffolds that have been recently described as possessing InhA inhibition activity [5,6,7,8]. Taking into consideration the increasing drug resistance toward available anti-TB drugs, the *Mycobacterium tuberculosis* ACP ⅽarrier proteiո (InhA) is a well-validated target. Different chemical classes have been reported to interfere with InhA. Rožman et al. [7] characterized four classes of InhA inhibitors (triclosan derivatives, pyridomycin, 4-hidroxy-2-pyridones, and thiadiazoles) and provided detailed information about their target engagement and in vivo activity. Prasad et al. [6] analyzed the structure–activity relationships of different scaffolds as key InhA inhibitors for anti-tubercular drug discovery. Sharma et al. [9] studied the antimicrobial, antiviral, and anti-TB effects of variety of hydrazone derivatives.

Isoniazid, also known as isonicotinic acid hydrazide (INH), demonstrates significant bactericidal activity and has become a key part of first-line anti-TB treatment courses [7]. INH is a prodrug that is activated by the enzyme catalase-peroxidase encoded by the KatG gene. Activated INH blocks the synthesis of essential mycolic acids by inhibiting *inhA*-encoded NADH-enoyl-ACP-reductase [10]. Ethionamide (ETH) is another important anti-TB prodrug, which is activated by enzyme EthA, a monooxygenase that is controlled by transcriptional repressor EthR. The activated ETH binds NAD+ to form the ETH-NAD complex, thus leading to the inhibition of InhA [6].

The dependency on KatG activation for INH-mediated killing is also the source of the main clinical weakness associated with the use of INH, because 40–95% of INH-resistant *M. tuberculosis* clinical isolates have katG mutations, leading to the reduced activation of INH (Figure 2) [7].

Therefore, the identification of direct InhA inhibitors—i.e., those that bind directly to InhA without the requirement of activation by KatG—is a reasonable strategy for overcoming INH resistance [11]. Some direct InhA inhibitors exhibit good affinity of the enzyme but show poor anti-TB activity. Further efforts should be focused on improving the membrane permeability and the bioavailability of these inhibitors by reasonable chemical modification.

Molecules carrying hydrazone functionality (the presence of azomethine group R^1^-HN-N=CR^2^R^3^) are of considerable interest due to their frequently reported anti-TB activities [12]. Furthermore, hydrazone’s inhibition capacity against other infectious pathogens makes its functionality an excellent fragment of newly designed anti-infective compounds such as antibacterial, antimalarial, antifungal, and anti-tubercular compounds [13].

Therefore, the aim of this review is a brief evaluation of the different mechanisms of action of available and recently approved anti-TB drugs and molecules in clinical trials. In addition, we describe our previous results on the synthesis and biological activity of hydrazide-hydrazones and thiadiazoles. Moreover, we summarize the data about the most-active synthesized compounds—those possessing InhA-inhibition activities.

## 2. Drug-Resistant Tuberculosis and Mechanism of Action of Currently Available Anti-Tuberculosis Agents

TB can be categorized into one of three main types, depending on the drug resistance of the infecting strain: drug-susceptible (DS-TB), multidrug-resistant (MDR-TB), and extensively drug-resistant (XDR-TB) [11]. DS-TB is the most common form of TB globally [14]. MDR-TB is caused by strains that are resistant to at least rifampin and isoniazid. Treatment of MDR-TB requires the use of second-line drugs such as aminoglycosides, cycloserine, and fluoroquinolones, and, less frequently, para-aminosalicylic acid, thioamides, or cyclopeptides. MDR-TB strains can also develop resistance to second-line drugs, thus making the situation even more complicated [14]. In 2006, XDR-TB was defined as MDR-TB that is additionally resistant to at least one of the fluoroquinolones and one of three injectable drugs (amikacin, kanamycin, and capreomycin) [14]. The term “totally drug-resistant tuberculosis” (TDR-TB) has been recently coined to describe infections by a strain that is resistant to all first-line and second-line anti-tubercular drugs [15]. A brief overview of the mechanisms of action of anti-tubercular drugs is presented in Figure 3 [16,17].

### 2.1. Cell Wall Synthesis Inhibitors

#### 2.1.1. Drugs Affecting Mycolic Acid (MA) Synthesis

The nitroimidazole class is represented by delamanid (Dlm), which was approved by the European Medicines Agency (EMA) in April 2014. Pretomanid is a nitro-imidazo-oxazin that blocks mycobacterial cell wall production and increases the release of nitric oxide. It was recently approved by the US Food and Drug Administration (US FDA) [18]. It is used for the treatment of adults with pulmonary XDR-TB via a combination of bedaquiline (BDQ) and linezolid_(LZD) [19].

InhA inhibitors inhibit the enzyme trans enoyl-acyl carrier protein (ACP) reductase, which plays an important role in the synthesis of MA. Based on their mechanism of action, they are mainly classified as either direct or indirect inhibitors [6]. Direct inhibitors are compounds that directly inhibit InhA without requiring any prior activation by Kat G and that are more favorable as novel drugs to fight MDR-TB [6]. Direct inhibitors include substances such as triclosan/diphenyl ethers, pyrrolidine carboxamides, pyrroles, acetamides, thiadiazoles, triazoles, methylthiazoles, hydrazones, and thiourea-based compounds [6,16]. Indirect inhibitors are compounds that require activation by Kat G to bind InhA and consequently block mycolic acid biosynthesis. These are drugs such as isoniazid (INH), ethionamide, and protionamide [6].

#### 2.1.2. Drugs Affecting Peptidoglycan Synthesis

Cycloserine inhibits two enzymes, L-alanine racemase and D-alanine:D-alanine ligase, thus impairing the peptidoglycan formation that is required for bacterial cell wall biosynthesis [20].

#### 2.1.3. Drugs Affecting Arabinogalactan Synthesis

Ethambutol (EMB) inhibits the polymerization of the cell wall arabinan of arabinogalactan and lipoarabinomannan by blocking arabinosyl transferases and induces the accumulation of D-arabinofuranosyl-P-decaprenol, an intermediate in arabinan biosynthesis. This results in halting bacterial growth [21].

### 2.2. ATP-Synthase Inhibitors

The diarylquinoline bedaquiline (Bdq) received marketing authorization, after accelerated and conditional approval, from the US FDA in December 2012 and the EMA in March 2014 [18]. Bdq acts by a novel mechanism of mycobacterial adenosine 5′-triphosphate (ATP)-synthase inhibition, with good clinical efficacy against strains that are resistant to multiple anti-TB drugs. However, Bdq was approved with a black box warning that highlighted the possibility of severe cardiovascular side effects [22]. The WHO first recommended the drug in June 2013 for the treatment of some forms of rifampicin-resistant and multidrug-resistant tuberculosis (RR/MDR-TB) in adults, while in October 2014, the WHO recommended a second new anti-tuberculosis agent, delamanid (Dlm), to treat RR/MDR-TB [18].

### 2.3. Inhibitors of Synthesis of DNA Precursors and DNA Gyrase

Para-aminosalicylic acid (PAS) competes with PABA for enzymes involved in folate synthesis, thereby suppressing the growth and reproduction of *M. tuberculosis* [23], while the fluoroquinolones (oxifloxacin and the more recently developed and more efficient moxifloxacin and levofloxacin) inhibit DNA gyrase.

### 2.4. Protein Synthesis Inhibitors

Rifampicin or rifampin (RIF) is a semi-synthetic derivative of rifamycin. It was first introduced as a first-line drug against tuberculosis in the 1970s and is currently still considered to be one of the most potent drugs in TB treatment [10]. The RNA polymerase beta-subunit is the main target of RIF that inhibits the transcription of messenger RNA [10]. Other agents with the same mechanism of action (but different targets in bacterial cells) are the cyclic peptide capreomycin and the aminoglycoside antibiotics kanamycin, streptomycin, and amikacin that act on the ribosomal complex (the small subunit RNA and the ribosomal protein RpsL).

### 2.5. Drugs with Another Mechanism of Action

Basically, pyrazinamide (PZA) is a prodrug that is activated by pyrazinamidase (encoded by the pncA gene) under acidic pH; thus, pncA mutations mediate PZA resistance in *M. tuberculosis* [10]. However, some clinical PZA-resistant strains with positive PZase do not have pncA gene mutations, suggesting the existence of alternative mechanisms.

Very recently, it was found that an ATP-dependent ATPase (ClpC1) involved in protein degradation and a bifunctional enzyme Rv2783 involved in the metabolism of RNA also confer PZA resistance in clinical isolates. These findings suggested that mutations of rpsA, panD, and clpC1 are involved in PZA resistance. Nevertheless, the clinical resistance of PZA in *M. tuberculosis* still needs further investigation [10].

## 3. Anti-Tubercular Structures Targeting InhA

### 3.1. InhA and Its Role in M. tuberculosis

In the search for a new potential mechanism for the treatment of TB and in light of continuous resistance to currently available drugs, the enzyme InhA reductase has become of interest because of its function and potential inhibition. InhA plays an essential role in the synthesis of fatty acid synthase two (FASII) and mycolic acid [24]. InhA catalyzes the NADH-dependent reduction of 2-trans-enoyl-ACP to yield NAD+ and a reduced enoyl thioester-ACP substrate, which, in turn, aids in the synthesis of mycolic acid [25]. Mycolic acids, which are unusually long-chain a-alkyl, b-hydroxy fatty acids containing 60–90 carbon atoms, are major components of the cell wall of *M. tuberculosis* [24]. Many antibiotic-resistant clinical tubercle bacillus isolates are resistant to the first-line anti-TB drug isoniazid. By inhibiting a 2-trans-enoyl-acyl carrier protein reductase, the InhA, this antibiotic has been shown to function on *M. tuberculosis*. However, the precise function of InhA in mycobacteria has remained uncertain [22]. Marrakchi et al. [26] isolateԁ a mycobacterial enzyme fraction containing InhA that displayed a long-chain fatty acid elongation activity with the characteristic properties described for the FAS-II system. The inhibition of this operation by inhibitors of InhA—isoniazid, hexadecynoyl-CoA, or octadecynoyl-CoA—has shown that InhA belongs to the system of FAS-II. In addition, the biosynthetic pathway of mycolic acids, which are the main lipids of the mycobacterial envelope, was also inhibited by the InhA inhibitors [24].

InhA is the only target of ethionamide (ETH) and protionamide (PTH) and is one of the targets of INH [4]. Mutations in InhA (rv1484) could induce *M. tuberculosis* resistance to ETH, PTH, and INH. It has been confirmed that less than 10% of INH-resistant *M. tuberculosis* clinical isolates harbor InhA mutations [4].

### 3.2. Hydrazide-Hydrazones

The inhibition capacity of hydrazone against other infectious pathogens makes hydrazone functionality an excellent fragment of newly designed antimycobacterial compounds [27]. As the hydrazide-hydrazone chemical scaffold (Figure 4) potentially defines promising candidates for the development of new potent anti-TB drugs, our studies were focused on their synthesis and characterization as potential InhA inhibitors.

In 2017, Angelova et al. [28] synthesized a series of new hybrid hydrazones bearing 2H-chromene or coumarin moieties. Twenty-two hydrazones were tested for their in vitro antimycobacterial activity against *M. tuberculosis* H37Rv and compared to the first-line drugs isoniazid and ethambutol. The citotoxicity against human embryonal kidney cell line HEK-293T was also evaluated. Among the tested hydrazones from the 2*H*-chromene series, compound **1** [28] (Figure 5) has proven to be the most effective compound against *M. tuberculosis* H37Rv (MIC = 0.13 µM), which is about eleven-fold more active than INH (MIC = 1.45 µM). Its antimycobacterial and cytotoxicity equieffective concentrations vary 700-fold, indicating a favorable selectivity index. From the 2-methyl-2H-chromene-bearing series, the most active compound with isoniazide fragment **2** [28] demonstrated MIC = 0.17 µM, which is about 10-fold more active than INH (MIC = 1.45 µM).

Additionally, we synthesized new coumarin-containing hydrazide-hydrazone derivatives and 2-aroyl-[1]benzopyrano [4,3-c]pyrazol-4(1H)-one analogs [29]. The antimycobacterial activity against the reference *M. tuberculosis* H37Rv strain and the cytotoxicity against the human embryonic kidney cell line HEK-293 were evaluated. The attained MICs were in the interval of 0.28–1.69 µM, which was comparable to the intervals of isoniazid and displayed good SI values ranging from 33 to more than 645. The most active compound **3** [29] (Figure 5) possessed potent antimycobacterial activity with MIC = 0.32 µM, combined with IC_50_ > 200 µM, which resulted in SI values higher than 625 [29]. The compound appears to be about 2.5-fold more active than INH (MIC = 0.79 µM). The results from the docking study indicate that the coumarin structure or the pyrazole-fused coumarin scaffold plays a very important role in the binding of moleculein the InhA pocket and is crucial to attaining favorable MIC values, whereas the substituents in the benzene ring were a key group for enhancing MIC values.

In addition, two new series of substituted indole- and indazole-based hydrazide-hydrazones were studied by Angelova et al. [30]. All compounds demonstrated significant MICs ranging from 0.39 to 2.91 µM. The cytotoxicity against the human embryonic kidney cell line HEK-293 was also evaluated and the selectivity (SI) of the antiproliferative effects was assessed. All compounds displayed good SI values ranging from >1978.83 to 12.04. The most active compounds **4** [30] (MIC = 0.4412 μM) and **5** [30] (MIC = 0.3969 μM) (Figure 5) demonstrated excellent antimycobacterial activity (2.1- and 2.3-fold more activity, respectively, than that of INH), a very low toxicity against the human embryonic kidney cell line HEK-293T, and high selectivity index values (SI = 633.49 and SI > 1978.83, respectively) [30]. The probability of the most promising antimicrobial compounds to inhibit the binding cavity of InhA was studied theoretically via molecular docking. The results revealed good docking and the importance of the 1,2,3-thiadiazole moiety in the connecting side chain.

As a continuation of a previous experiment, Angelova and Simeonova [32] chose compound **5** [30] (MIC = 0.3969 μM) (Figure 5) to examine further its effects on the liver and kidney functions of female mice. For a period of 14 days, the compound was administered orally in doses of 100, 200, and 400 mg/kg bw, while orally using INH as the control at a dose of 50 mg/kg bw It appeared that compound **5** [30], unlike INH in the control mice, did not affect the urine and serum hematological and biochemical parameters, did not significantly affect the quantity of MDA, and sustained its level near to the control values, though lower by 36% (*p* < 0.05) than in the INH-treated animals.

In continuation to the hydrazide-hydrazones work, in 2022, Angelova et al. [31] synthesized two series of hydrazide-hydrazone and sulfonyl hydrazone derivatives. Their antimycobacterial activity was assessed using *Mycobacterium tuberculosis* strain H37Rv and all compounds demonstrated significant MICs from 0.07 to 0.32 µM, which were comparable to those of isoniazid. Human embryonic kidney cells HEK-293T and mouse fibroblast cell line CCL-1 were used for cytotoxicity assessment. The lead derivative from the hydrazide-hydrazone series, compound **6** [31], bearing a thiadiazole scaffold (Figure 5), demonstrated the highest antimycobacterial activity (MIC = 0.0730 µM), which was around two-fold less active than INH (MIC = 0.0343 µM) and minimally associated cytotoxicity against the abovementioned two normal cell lines (SI = 3516, HEK-293 and SI = 2979, CCL-1). From the sulfonyl hydrazone series, compound **7** [31] (Figure 5) showed MIC 0.0716 µM, about two -old less activity than that of INH (MIC = 0.0343 µM). The performed molecular docking studies on two crystallographic structures of enoyl-ACP reductase InhA demonstrated that there is a high chance of interaction between the prepared compounds and the InhA receptor.

Valcheva et al. [33] carried out an additional evaluation of compounds **4** [30] and **6** [31] (Figure 5), focusing on their acute and subacute toxicity in mice, their redox-modulating capacity by in vivo and in vitro investigations, pathomorphological observation in the liver, kidney, and small intestine tissue specimens, and intestinal permeability. A histological examination of the different tissue specimens did not reveal toxic changes. The modifications appeared to successfully improve the permeation profiles of INH derivatives. After the mice were treated for 14 days with both molecules, weight gain and absence of statistically significant changes in the hematological, biochemical, and pathomorphological parameters in the blood, livers, small intestines, and kidneys were observed, providing evidence for good tolerability toward the examined derivatives.

In addition to the studies by our scientific team, our review of available recent literature led to the discovery of other described hydrazide-hydrazone compounds with InhA inhibition activity, which are summarized below.

Naveen Kumar et al. [34] synthesized and characterized hydrophobic *N*-acylated isonicotinic acid hydrazide derivatives as potential InhA inhibitors and tested them against *Mycobacterium tuberculosis* H37Rv and two human clinical isolates by means of a colorimetric tetrazolium microplate assay. Compound **8** [34] (Figure 6) exhibited enhanced anti-TB activity with MIC = 0.096 µM against the *Mycobacterium tuberculosis* H37Rv strain and MIC—0.049 µM against both human clinical isolates *Mycobacterium tuberculosis*-1 and *Mycobacterium tuberculosis*-2; Log P—8.02 indicated that the compound is highly lipophilic; LD50 was estimated to be greater than 5000 mg/kg body weight for the compound. The compound was six-fold more active than INH, which was used as a positive control, with MIC = 0.57 µM.

Pahlavani et al. [35] described the synthesis of compound **9** (Figure 6) derived by the condensation of isonicotinoyl hydrazide and 3-ethoxysalicylaldehyde. The compound was subjected to antimicrobial and antitubercular activity screening. It showed MIC = 4 µg/mL (0.0140 µM) *Mycobacterium tuberculosis* H37Rv strain, which was 160-fold less active than isoniazid, which was used as control (MIC = 0.025 µg/mL). The new synthesized compound showed good biological activity against tested bacteria and high levels of activity against *Mycobacterium tuberculosis* (H37RV) in vitro.

In another study, a new series of indolylhydrazones were synthesized and screened for in vitro anti-tubercular activity [36]. The indolylhydrazone derivative **10** [36] (Figure 6) exhibited *Mycobacterium tuberculosis* MIC > 25 µg/mL (0.067 µM), which showed 100-fold weaker activity than that of RIF, which was used as positive control, with MIC = 25 µg/Ml.

In 2016, Velezheva et al. [37] prepared a series of novel hydrazide-hydrazones combining indole and pyridine nuclei and studied their in vitro antimycobacterial activity against H37Rv and against a clinical isolate of INH-resistant *M. tuberculosis* with selective single INH resistance, designated as CN-40. Compound **11** [37] (Figure 6) showed MIC = 0.05 µg/mL (1.43 µM) against H37Rv, which was approximately as active as INH against H37Rv (MIC = 0.06 µg/mL). However, unlike INH, compound **11** [37] showed appreciable activity against the INH-resistant *M. tuberculosis* (MIC = 2–5 µg/mL). The compound was tested for in vitro cytotoxicity to macrophages and showed IC_50_ = 15 and SI = 300, which defined it as about seven-fold less toxic than INH (IC_50_ > 100, SI > 1666).

Ghiano et al. [38] reported a library of fifteen *E,Z* pairs of *N*-substituted tosyl *N*’-acryl-hydrazones, which are structurally related to isoniazid. The collection was assayed against *M. tuberculosis*; the most active compounds **12**–**14** [38] (Figure 6) were *E* isomers, holding bulky aromatic (phenyl and naphthyl) substituents, while the acrylate group was critical for antimycobacterial activity. The compounds were tested against an INH overexpressing strain. The attained MIC differences provided evidence for the possibility of the compounds to inhibit the enzyme. The three compounds showed MIC against MtH37Rv 1.25 µM, which was 3.5-fold less active than INH (MIC = 0.36 µM) and four-fold less active than RIF (MIC = 0.30 µM), which were used as positive controls. It appeared that the three derivatives presented the same binding mode after they were docked on the crystallized InhA protein.

Santoso et al. [39] prepared three isoniazid-isatin hydrazone derivatives, investigated their in vitro anti-tubercular activity, and performed additional molecular docking. The highest anti-TB effect was observed in compound **15** [39] (Figure 6), with MIC of 0.017 mM (0.17 µM), in comparison with the positive control, RIF with MIC = 0.048 mM. Additionally, molecular docking studies were performed to examine further the interaction between the isoniazid-isatin hydrazone derivatives and the active site of the InhA receptor. The results confirmed the experimental data.

Koçak Aslan et al. [40] linked INH to in-house synthesized sulfonate esters via a hydrazone bridge and tested their inhibitory activity against InhA spectrophotometrically. The results of the study showed that the primary mechanism of anti-TB activity of the compounds is not related to direct InhA inhibition, even though most of them are recognized by the binding pocket of InhA. Compound **16** [40] (Figure 6) was found to be a moderate inhibitor of InhA, with 41% inhibition, compared to triclosan (98% inhibition), which indicated that the introduction of iodine as a bulky and lipophilic halogen can be an important modification to bind to the active site of the enzyme. The compound exhibited MIC against the *Mycobacterium tuberculosis* H37Rv strain of 0.62 µM, which was two-fold less activity than that of INH (MIC = 0.31 µM).

Fernandes et al. [41] reported the in vitro and in vivo evaluation of anti-TB activity against actively replicating and dormant *M. tuberculosis* H37Rv (MIC90) and cytotoxicity against the MRC-5 cell line of 22 new N-oxide-containing derivatives Demonstrating MIC90 values of 1.10 and 6.62 μM against active and nonreplicating *Mycobacterium tuberculosis*, respectively, compound **17** [41] appeared to be the leading molecule in the study. The safety and efficacy of compound **17** were further proved; the compound was orally bioavailable and highly effective, and it reduced significantly the levels in a mouse model of infection. The conducted microarray-based initial studies suggested that the compound acts through the blocking of the translation.

In another study by Kumar et al. [42], isonicotinoyl hydrazone derivatives were prepared and examined for in vitro antimycobacterial activity against *M. tuberculosis* H37Rv and two clinical isolates by means of a tetrazolium microplate assay (TEMA). Among the tested compounds, **18** [42] (Figure 7) was the most potent, presenting an inhibition concentration at 0.59 µM, compared to isoniazid, which was used as control, at 0.57 µM.

More et al. [43] prepared 52 novel pyrrole hydrazine derivatives, which targeted the essential enoyl-ACP reductase. During the performed binding model, it was suggested that one or two hydrogen bonding interactions were formed between the pyrrole hydrazones and the InhA enzyme. The lead compound **19** [43] (Figure 7) with MIC 0.2 µg/mL (4.86 µM) occupied the same binding site as that of PT70 and TCL.

Ghiya and Joshi [44] prepared a series of *N*’-substituted-4-methylbenzenesulfonohydrazide derivatives that were synthesized and evaluated for antimycobacterial activity. Compound **20** [44] appeared to have excellent inhibition activity with MIC = 48.04 µM, compared to INH, which showed MIC of 3.6 µM.

Joshi et al. [45] identified novel hydrazone ligands and their copper complexes as potent InhA inhibitors. The conducted studies showed that ligands are less active compared to metal complexes. Among the complexes, **21** [45] (Figure 7) showed MIC = 0.8 µg/mL (9.43 µM) against mycobacteria, with no apparent cytotoxicity toward the human lung cancer cell-line (A549). During the in vitro InhA inhibition assay that was performed with triclosan as control, the copper complex **21** [45] demonstrated 100% binding with the enzyme, with IC50 at 2.4 µM, whereas its corresponding ligand showed the same enzymatic activity and IC50 at 7.7 µM.

A series of new organometallic tosyl hydrazones were reported by Concha et al. [46]. Their anti-tubercular activity evaluation, measured in vitro against the *Mycobacterium tuberculosis* mc26230 strain, revealed that all organometallic tosyl hydrazones were considerably less active than the reference drug isoniazid. Compound **22** [46] (Figure 7) showed MIC of 183 µM, compared to 0.4 µM of INH.

Oliveira et al. [47] reported a series of isoniazid derivatives bearing a phenolic or heteroaromatic coupled frame, which were obtained by mechanochemical means, and their activity against *Mycobacterium tuberculosis* cell growth was evaluated. Compound **23** [47] (Figure 7) showed MIC of 0.11 µM, compared to INH, with 0.36 µM. As an InhA inhibition assay was performed, compound **23** showed 19% inhibition at 50 µM, in comparison to triclosan >99%.

A variety of pyrazine derivatives, which contain hydrazide-hydrazone moiety, were reported by Hassan et al. [48], who additionally assessed their antimycobacterial activity against *Mycobacterium tuberculosis* H37Rv. From the prepared thirty-one derivatives, compound **24** [48] demonstrated the lowest value of MIC, at 0.78 µg/mL, which equals 0.0017 µM. This was similar activity to that of INH (0.1 µg/mL), approximately two-fold more activity than that of ETH (1.56 µg/mL), and eight times more activity than that of PZA (6.25 µg/mL). Additionally, during the cytotoxicity study, the compound showed no cytotoxicity against peripheral blood mononuclear cells (PBMC), with IC50 = 846.9 µg/mL and high SI = 1085.7. The methods of pharmacophore mapping and inverse molecular docking were used to identify the target of the lead compound. The obtained data suggested that the possible target of compound **24** was pantothenate synthetase.

In 2018, in a continuation of an earlier study [50], in which a series of hydrazones active against *M. tuberculosis* were identified, Bonnett et al. [49] selected five compounds for additional analysis. Compound **25** [49] (Figure 7), possessed hydrazide-hydrazone moiety and exhibited a MIC value of 14 ± 7 µM against the *Mycobacterium tuberculosis* reference strain. In the later study, the compound was additionally examined in both aerobic and anaerobic conditions using the low-oxygen-recovery assay (LORA). This resulted in IC90 values of 22 ± 12 µM in anaerobic conditions and 6.4 ± 2.4 µM in aerobic conditions.

Nogueira et al. [51] reported a series of vitamin B6-containing hydrazones and *N*-acylhydrazones that were examined against *Mycobacterium tuberculosis*. From the *N*-acylhydrazones, the leading compound **26** [51] (Figure 8) had MIC = 10.90 µM, while derivative **27** [51] showed MIC = 72.72 µM, which was the best result in the hydrazone series.

Sampiron et al. [52] assessed the potential of three benzohydrazones, four isoniazid-acylhydrazones, and one hydrazone as anti-tuberculosis agents. The INH-acylhydrazone **28** [52] (Figure 8) showed the lowest MIC = 0.12 µg/mL (4.98 µM) among all the compounds tested in that research.

Rohane et al. [53] reported a study, that described the design of 51 hydrazone derivatives of eugenol and their docking with two enzymes of the H37Rv strain. Ten hydrazone derivatives of eugenol were synthesized and tested for their anti-tubercular activity. All compounds appeared to have significant activity against H37Rv at concentrations of 50 and 100 µg/mL, whereas compound **29** [53] (Figure 8) the highest activity at 25 µg/mL (0.068 µM). In comparison, INH was active at 12.5 µg/mL (0.091 µM).

The aAnti-tubercular potential against *Mycobacterium tuberculosis* reference strain H37Rv of eleven quinoline hydrazone derivatives was examined in a study by Sruthi et al. [54]. Compounds **30**–**32** [54] (Figure 8) appeared to possess promising potential with an MIC value of 4 µg/mL against *Mycobacterium tuberculosis* reference strain H37Rv, using the broth microdilution method, in comparison to RIF, which was used as a control substance and showed an MIC value of 0.025 µg/mL (3.038 µM). The cytotoxicity of the three compounds was also evaluated against the HepG2 cell line by an MTT cytotoxicity assay. The obtained data indicated that the compounds are highly selective against *Mycobacterium tuberculosis*, as the IC50 values were found to be >100 µM.

The object of a study of Padmini et al. [55] was the preparation of new pyrazole acetamides that possess a hydrazone group and the assessment of their anti-tubercular activity. The results were that compound **33** [55] (Figure 8) demonstrated an MIC value of 3.12 μg/mL (0.01 µM) against *Mycobacterium tuberculosis*H37Rv. Moreover, the presence of important hydrogen bonding interactions with InhA was proven by the docking studies.

Two series of alkyl hydrazides and hydrazones were described by Faria et al. [56], with promising in silico properties such as membrane permeabilities and spontaneous IN* radical formation. In the case of the hydrazone series, molecules **34** [56] and **35** [56] (Figure 8) presented MIC values against the *Mycobacterium tuberculosis* reference strain of 0.3 µM, which were similar to that of INH (also 0.3 µM). They also showed MIC values > 128 and 128 µM, respectively, against H37RvINH (ΔkatG), a strain with full deletion of the KatG gene. In comparison, INH exhibited a value of 933.4 µM.

Karunanidhi et al. [57] reported novel isatin hydrazones and their thiomorpholine-tethered analogs. During the initial screening for anti-mycobacterial activity against the H37Rv strain of *Mycobacterium tuberculosis* under level-I testing, compound **36** [57] (Figure 9) showed highest activity, with IC50 = 1.9 µM. Following the level-II testing against the five drug-resistant strains—isoniazid-resistant strains (INHR1 and INH-R2), rifampicin-resistant strains (RIF-R1 and RIF-R2), and fluoroquinolone-resistant strain (FQ-R1)—**36** demonstrated IC50 of 3.6 µM against the RIFR1 *Mycobacterium tuberculosis* strain, followed by the INH-R1 *Mycobacterium tuberculosis* strain with IC50 of 3.5 µM and IC50 5.9 µM against FQ-R1.

Pflégr et al. [58] designed twenty new 2-(2-isonicotinoylhydrazineylidene)propanamides assayed against susceptible *Mycobacterium tuberculosis* H37Rv, nontuberculous mycobacteria, and the MDR-TB strain. Molecule **37** [58] (Figure 9) appeared to be the leading compound in the study with MIC values of 0.03 µM against the H37Rv strain and it was used to determine the mechanism of action, which resulted in a decrease in the mycolic acids production. It showed MIC = 6 µM against InhA-overproducing strain and, in contrast, MIC = 0.15–0.3 µM against the KatG overproducing strain, which indicated that the compound acts by InhA inhibition.

Desale et al. [59] prepared and characterized nine hydrazide-hydrazone derivatives, which were further tested for their in vitro anti-tubercular activity. Except for two derivatives, all other hydrazones retained strong potency. Compound **38** [59] (Figure 9) exhibited a higher potency of 1.6 µg/mL (0.006 µM), in comparison with the controls, pyrazinamide (MIC: 3.12 µg/mL 0.025 µM), ciprofloxacin (MIC: 3.12 µg/mL 0.009 µM), and streptomycin (MIC: 6.25 µg/mL 0.01 µM). Additionally, molecular docking analysis against the acyl-CoA carboxylase showed that compound **38** [59] had the highest docking score of −9.65 Kcal/mol, in comparison with the used standards ciprofloxacin (−3.57 Kcal/mol), streptomycin (−2.82 Kcal/mol), and isoniazid (−5.65 Kcal/mol). This suggested that the probable targeting method for the synthesized compounds was targeting the mycolic acid biosynthesis pathways, thereby interfering with the cell envelope integrity of *M. tuberculosis*.

Gobis et al. [60] prepared and evaluated the tuberculostatic activity of four novel methyl 4-phenylpicolinoimidate derivatives of hydrazone. Their tuberculostatic activity was examined by means of the standard strain H37Rv and two native strains from patients, Spec 210 (which is resistant to clinically used anti-tuberculosis drugs) and Spec 192 (which is completely sensitive). Derivatives containing nitrofuran systems demonstrated higher tuberculostatic activity. The lead compound **39** [60] (Figure 9) showed an MIC of 3.1 µg/mL (0.009 µM) against both sensitive and resistant strains and appeared to be four times more active than the reference INH.

Briffotaux et al. [61] performed a whole-cell-based screening for compounds with antimycobacterial activity in the presence of linezolid. Compound **40** [61] (Figure 9) appeared to be a promising anti-TB agent, being a small molecule consisting of an adamantane moiety and a hydrazide-hydrazone moiety. The compound was suggested as a new inhibitor of MmpL3 because it binds to the same pocket as other already-published MmpL3 inhibitors. It showed an MIC of 0.2 µg/mL (6.66 µM) against *Mycobacterium tuberculosis* H37Rv and appeared to be sensitive against a panel of sensitive rifampicin-resistant and MDR strains clinical isolates.

As potential anti-TB agents, Akki et al. [62] prepared a series of new coumarin hydrazone oxime scaffolds and tested their in vitro activity against the *Mycobacterium tuberculosis* H37Rv strain, and Vero cells were used to assess cytotoxicity. The lead candidate—compound **41** [62] (Figure 9)—exhibited an MIC of 0.78 μg/mL (0.021 μM), demonstrating more potent anti-TB activity than rifamycin and a comparable activity to that of INH. Minimal cytotoxicity against Vero cells was observed, indicating a good safety profile.

Abdelhamid et al. [63] reported three sets of 4,4-diphenyl-2-hydrazinyl-1*H*-imidazol-5(4*H*)-one hydrazones. During in vitro screening against *Mycobacterium smegmatis* ATCC 607, two leading compounds were discovered, **42** [63] with MIC = 0.033 µM. and **43** [63] (Figure 9) with MIC = 0.032 µM. Molecule 43 showed a bactericidal effect comparable to that of INH. Their cytotoxicity assessment against the Vero cells line showed that they had minimal in vitro toxicity with good selectivity indices. Molecular docking studies were performed on InhA and revealed that the enzyme is a possible target for their anti-tubercular activity. The two compounds demonstrated the highest docking score values of S = −14.09 and −15.13 Kcal/mol, respectively, which were in alignment with the antimycobacterial activity.

Senthilkumar et al. [64] described eight new (*E*)-4-amino-N’-(substituted benzylidene) benzohydrazides, which were assessed for their antimicrobial and antifungal activity. Furthermore, molecular docking studies were performed with them, using InhA protein as a potential target of action. The results indicated the highest glide score of—7.298 in compound **44** [64] (Figure 9), due to the H bond with water and SER 94.

In 2023, Lone et al. [65] studied 14 derivatives of isonicotinoylhydrazonobutanoic acid. The best dock score was observed in compound **45** [65] (Figure 9) during the molecular docking studies. Their antimycobacterial activity was examined against the following M.Tb strains: H37Ra, H37Rv and three clinical isolates of INH-resistant strains (INH-R D, INH-R G, and INH-R K). This showed MIC = 1 µg/mL (0.0042 µM) against both the H37Ra and H37Rv strains. Moderate activity against all three INH-R strains (D, G, and K), with an MIC value of 64.µg/mL, was observed. The cytotoxicity assay of the compound resulted in low toxicity against three normal human cell lines (HEK-293, AML-12, and RAW-264), up to a 500 µM concentration.

### 3.3. Thiadiazoles

Thiadiazoles occur naturally in four different isomeric forms (1,2,3-thiadiazole, 1,3,4-thiadiazole, 1,2,4-thiadiazole, and 1,2,5-thiadiazole), having one sulfur and two nitrogen atoms with a hydrogen-binding domain, as presented in Figure 10 [66,67].

Thiadiazole-based InhA inhibitors were initially discovered in a high throughput screening campaign by GlaxoSmithKline Pharmaceuticals, Ltd. [68,69]. Preliminary structure–activity relationship studies were later performed by AstraZeneca, where it was also demonstrated that this binding is dependent on the oxidation state of NADH [8]. The methyl thiazole scaffold **46** [68,69] (Figure 11) was reported as a direct InhA inhibitor, where potent enzyme inhibition (InhA IC_50_ = 0.003 μM) translates into cellular potency (*Mycobacterium tuberculosis* MIC = 0.19 μM) [8].

Therefore, in 2013, Shirude et al. [8] described new analogs of the methyl thiazole series 46 [68,69], which acted through a similar mode of inhibition to the reported methyl thiazoles while showing improved cytochrome P450 (CYP) inhibition and a safety profile. Compound **47** [8] (Figure 11) with the open amide appeared to be the new lead. The crystal structure of InhA in combination with **47** [8] revealed similar interactions to that of **46** [68,69]. The compound was tested for InhA enzyme inhibition in a fluorescence-based assay and showed InhA IC_50_ = 0.2 µM. The MIC against *Mycobacterium tuberculosis* H37Rv strain was 3.75 µM. During the study, a novel mechanism of InhA inhibition was also identified and characterized by a hitherto unreported “Y158-out” inhibitor-bound conformation of the protein that accommodates a neutrally charged “warhead”.

Based on the tetracyclic thiadiazole structures, Šink et al. [15] designed a series of 23 truncated analogues that had only three aromatic rings. They aimed to preserve the thiadiazole and thiazole rings, while the other side of the molecule was replaced by different alternative groups. Their InhA inhibition activity was measured fluorometrically, and then this effect was translated into investigating their antimycobacterial activity in terms of MICs. The study discovered that the lead compound **48** [15] (Figure 11) was the most potent InhA inhibitor (IC_50_ = 13 nM), as well as the best antimycobacterial compound (MIC = 2 µM) in this series of truncated analogs.

In a study by Joshi et al. [70], a drug-design approach was employed to generate new structures based on the combination of molecular docking and 3D-QSAR studies on the novel derivatives of pyrrole-containing aryloxy thiadiazole that were explored as novel anti-TB agents. Among the 68 investigated compounds, **49** [70] (Figure 11) displayed significant activity (3.125 µg/mL or 0.009 µM) against the *M. tuberculosis* H37Rv strain. In addition, potential toxicity toward mammalian Vero cell lines and A549 (lung adenocarcinoma) cell lines was evaluated, where compound **49** [70] showed IC_50_ values of 220 ± 0.3 µM and 225 ± 0.3 µM, respectively, which was around two-fold less toxic compared with standard INH. Docking analysis of the crystal structure of enoyl-acyl carrier protein reductase (ENR) indicated the occupation of pyrrolyl-substituted aryloxy 1,3,4-thiadiazole into the hydrophobic pocket of InhA enzyme. Docking studies indicated that compound **49** [70] showed that the orientation and conformation was similar to that of the crystallographic ligand pyrrolidine carboxamide.

Later, Karabanovich et al. [71] described the discovery and structure–activity relationships of nitro-substituted 2-Alkyl/Aryl-5-benzylsulfanyl-1,3,4-thiadiazoles. The investigated compounds had a highly selective antimycobacterial effect as it was low in vitro toxicities in four proliferating mammalian cell lines and in isolated primary human hepatocytes. Several in vitro genotoxicity assays indicated that the selected compounds had no mutagenic activity. Compound **50** [71] (Figure 11) exhibited an MIC against *Mycobacterium tuberculosis* (H37Rv) of 0.06 µM, which was about eight-fold more activity than that of INH (MIC = 0.5 µM). Furthermore, selected compounds bearing 3,5-dinitrobenzylsulfanyl groups from the series, including compound **50** [71], exhibited outstanding activities against six MDR strains of *Mycobacterium tuberculosis*, with MIC values of 0.25–0.5 μM and no cross-resistance with any first- or second-line anti-TB drugs. The SAR study determined that 3,5-dinitro substitution had a crucial role in antimycobacterial activity; any changes to the positions or numbers of the nitro groups led to a significant decrease in antimycobacterial activity.

Mali et al. [72] designed and synthesized a series of fatty-acid thiadiazole derivatives and established their in vitro antimycobacterial potential. Out of 22 fatty-acid thiadiazole derivatives, compound **51** [72] (Figure 12) was the most active, exhibiting an MIC of 2.34 μg/mL (0.0065 µM) against *Mycobacterium tuberculosis* H37R, which was about six-fold less activity than that of INH (MIC = 0.6 μg/mL or 0.044 µM) and four-fold less activity than that of RIF (MIC = 0.6 μg/mL or 7.3 µM). All the molecules were docked into the active site of the crystal structure of enoyl-ACP reductase, which revealed that compound **51** [72] binds to the enzyme InhA.

A series of substituted 1,3,4-thiadiazole derivatives were also synthesized by Patel et al. [73] and evaluated for their in vitro anti-mycobacterial activity against the *Mycobacterium tuberculosis* H37Rv and the MDR-TB strain. Among the compounds tested, compound **52** [73] (Figure 12) showed significant inhibitory activity with an MIC of 9.87 µM (H37Rv strain) and 9.87 µM (MDR-TB strain), compared to that of isoniazid [MIC of 3.64 µM (H37Rv) and >200 µM (MDR-TB strain)] and rifampin [MIC of 0.152 µM (H37Rv) and 128 µM (MDR-TB strain)]. The compounds were additionally evaluated for their cytotoxicity against the mammalian Vero cell line, which confirmed anti-mycobacterial activity at a non-cytotoxic level.

Doğan et al. [12] synthesized 18 new thiadiazolylhidrazones (TDHs), introducing two (substituted) phenyl rings to obtain better antimycobacterial activity. When these compounds were evaluated for their in vitro anti-tubercular activity against *M. tuberculosis* H37Rv, **53** [12] and **54** [12] (Figure 12) stood out as excellent anti-tubercular agents in the series with an MIC value of 0.78 mg/mL, showing 2.4 and 2.2 log reductions in growth during the *M. tuberculosis* nutrient starvation assay. They were evaluated for their in vitro inhibition of InhA from *M. tuberculosis* at 50 mM using triclosan as the positive control. Compound **53** [9] (Figure 12), which had the lowest MIC value from the microplate alamar blue assay (MABA), was the best InhA inhibitor with 75% inhibition of the enzyme. All tested compounds were found to be non-toxic.

Considering the results summarized in Table 1, the hydrazide-hydrazone and thiadiazole derivatives were promising lead compounds for the development of a novel chemical class of anti-tubercular drugs. The effect of the heterocyclic or aromatic substituents is shown in Table 1, which lists the anti-tubercular activity of the reported hydrazone derivatives against the *Mycobacterium tuberculosis* H37Rv strain (MIC), where the selectivity index (SI), LD50 (mg/kg bw), and InhA inhibition activity were tested.

Even though a comprehensive comparison is difficult to explain, we undertook a simple comparison to evaluate the effect of the existence of the heterocyclic ring moieties. Altogether, INH-acyl hydrazones containing pyridine fragments can be highlighted as preferred for the development of promising anti-TB drug candidates (Table 1, showing 16 lead compounds from 35 references for the hydrazone derivatives). In addition, the results against the standard strain *Mycobacterium tuberculosis* and clinical isolates of some molecules showed an elevated selectivity index (SI) and lower cytotoxicity and hepatotoxicity than INH. While pyridine substituent showed good anti-tubercular activity, the presence of a para-toluyl functional group on compound **45** enhanced the anti-tubercular activity to 0.004 µM as the MIC value against both H37Ra and H37Rv strains, which was higher than those ofall the piridine-based hydrazine–hydrazones [65].

As shown in Table 1, the hydrazide-hydrazones with a 1,2,3-Thiadiazole ring showed the highest anti-tubercular activity on compounds **5** and **6**, as tested by our investigated team, with a very good selectivity index (1979 and 3516, respectively) and LD50 > 2000 mg/kg bw.

Additionally, the review confirmed that 1,3,4-thiadiazole-based derivatives provided high anti-tubercular activity, due to the suitable interaction with InhA as the protein receptor. The effect of the other functional groups, especially the heterocyclic moiety, is shown in the table, as well as their interaction with the InhA receptor, which was evaluated through molecular docking studies.

From this report, it can be concluded that hydrazone derivatives with thiadiazole heterocyclic ring moieties are pivotal for anti-tubercular activity. A further heterocyclic hydrazone derivative design needs to consider those moieties to obtain a potential anti-tubercular agent against *Mycobacterium tuberculosis* H37Rv.

## 4. Conclusions

In view of the continuous emergence and spread of strains that are resistant to many or most of the currently available antimycobacterial drugs, the discovery of new potent and non-toxic anti-TB drug candidates is critical, especially taking into consideration the long-lasting consequences of the COVID-19 pandemic. INH remains among the most frequently prescribed drugs for treatment of TB disease, but resistance to the currently available drugs may reduce the successful outcome of the therapy.

Since the InhA enzyme has been widely reported to play a crucial role in the mycolic acid pathway and fatty acid biosynthesis, it has become of increased interest as a target for the development of new anti-TB drugs. Derivatives from different chemical classes were examined for their InhA inhibition. In the current review, a detailed observation was made on the recently reported hydrazide-hydrazone and thiadiazoles, which successfully form a complex with InhA, inhibiting its activity in the mycolic acid biosynthesis and, therefore, suppressing the growth of *M. tuberculosis*.

Our review highlighted that thiadiazoles and hydrazide-hydrazones are potent and non-toxic antimycobacterial agents with InhA inhibition activity and can be considered for further development.

## Figures and Tables

**Figure 1 pharmaceuticals-16-00484-f001:**
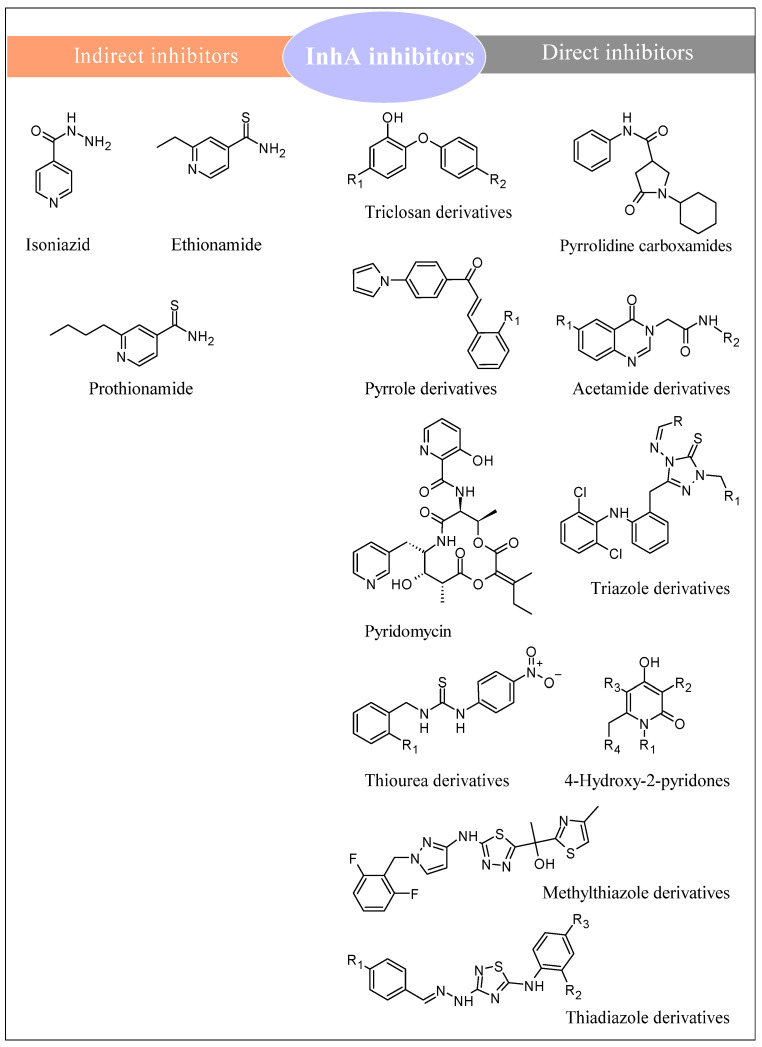
Direct and indirect InhA inhibitors, as described in the literature [5,6,7,8].

**Figure 2 pharmaceuticals-16-00484-f002:**
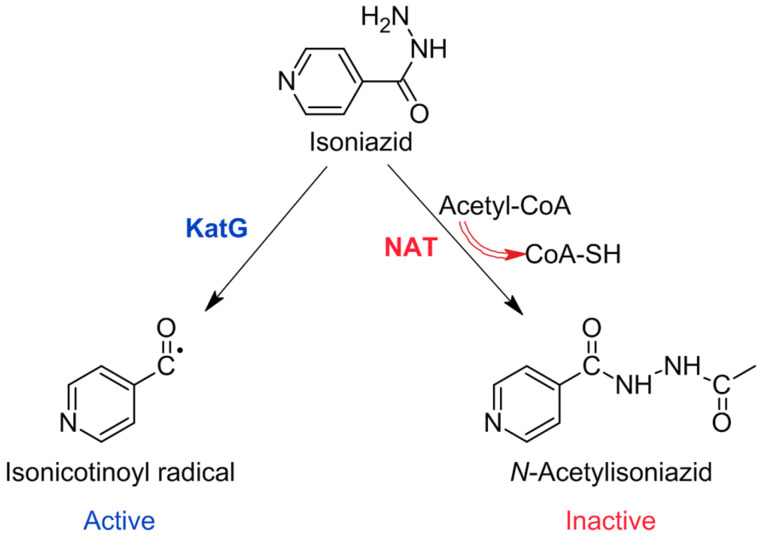
Activation and inactivation pathways of INH.

**Figure 3 pharmaceuticals-16-00484-f003:**
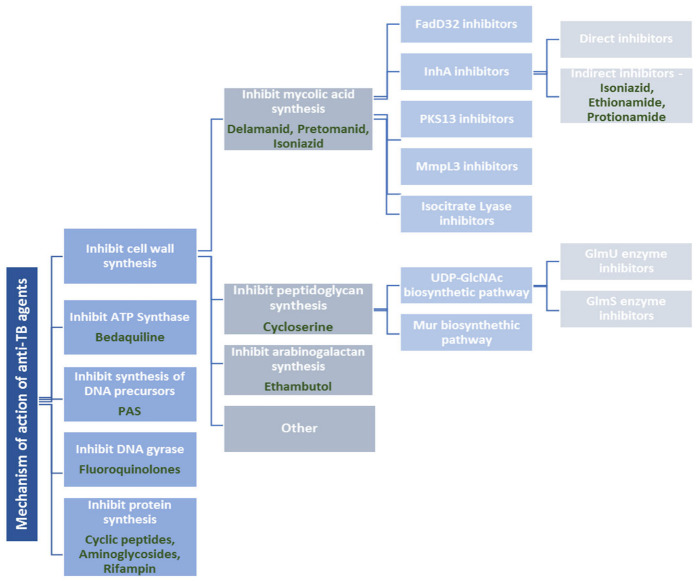
Mechanisms of action of the currently available anti-tuberculosis drugs [16,17].

**Figure 4 pharmaceuticals-16-00484-f004:**
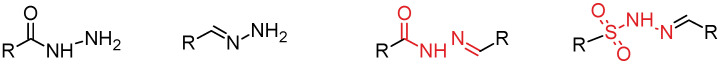
The structures carrying hydrazone functionality.

**Figure 5 pharmaceuticals-16-00484-f005:**
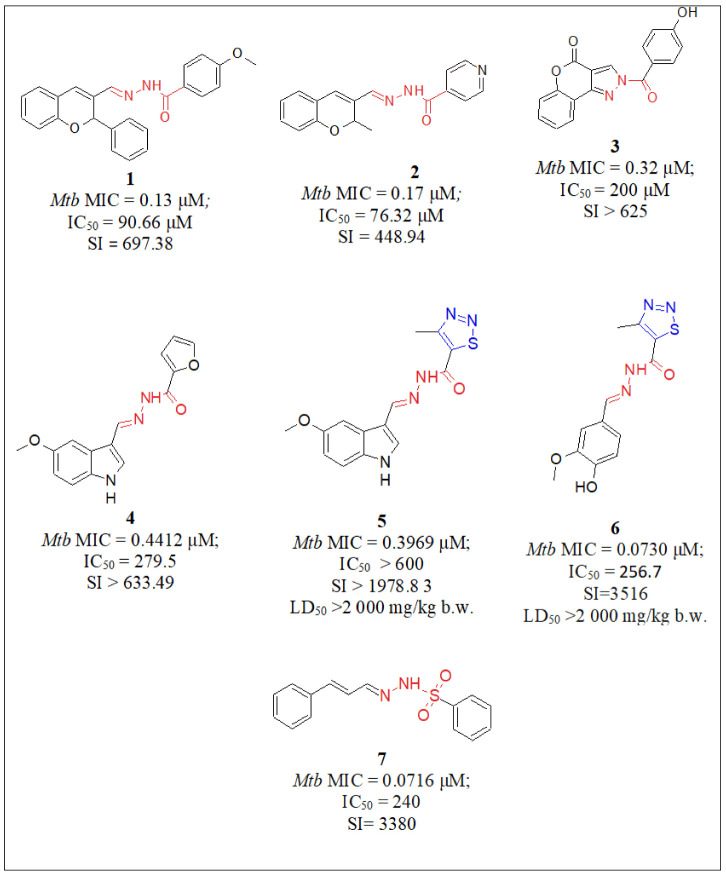
Chemical structures of compounds **1** [28], **2** [28], **3** [29,30], **4** [30], **5** [30], **6** [31] and **7** [31].

**Figure 6 pharmaceuticals-16-00484-f006:**
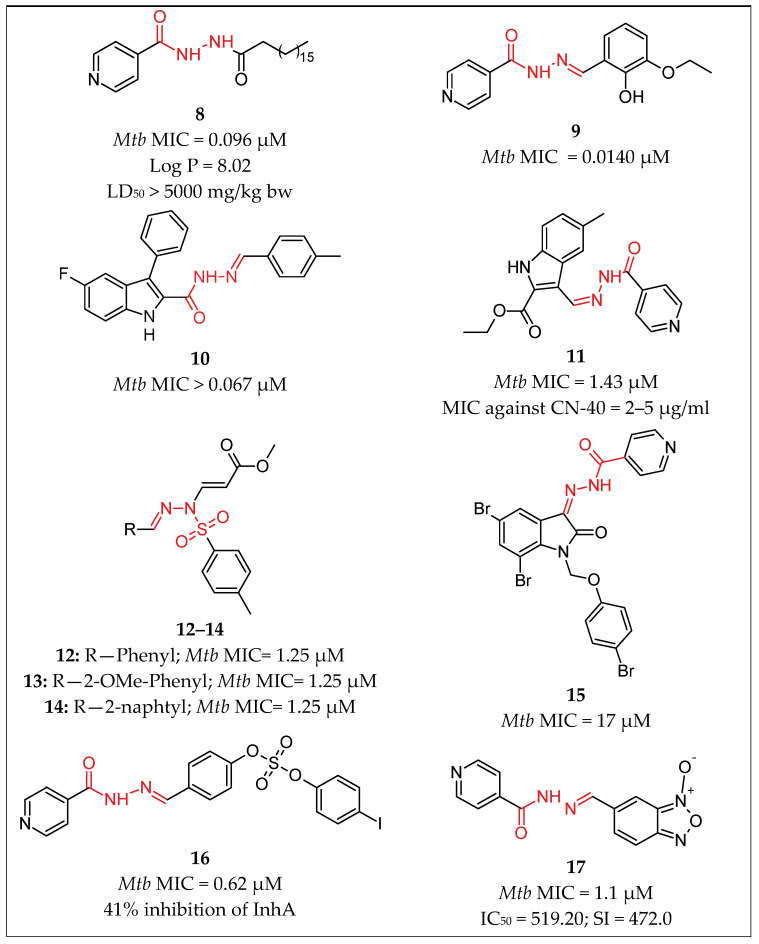
Chemical structures of compounds **8** [34], **9** [35], **10** [36], **11** [37], **12**–**14** [38], **15** [39], **16** [40] and **17** [41].

**Figure 7 pharmaceuticals-16-00484-f007:**
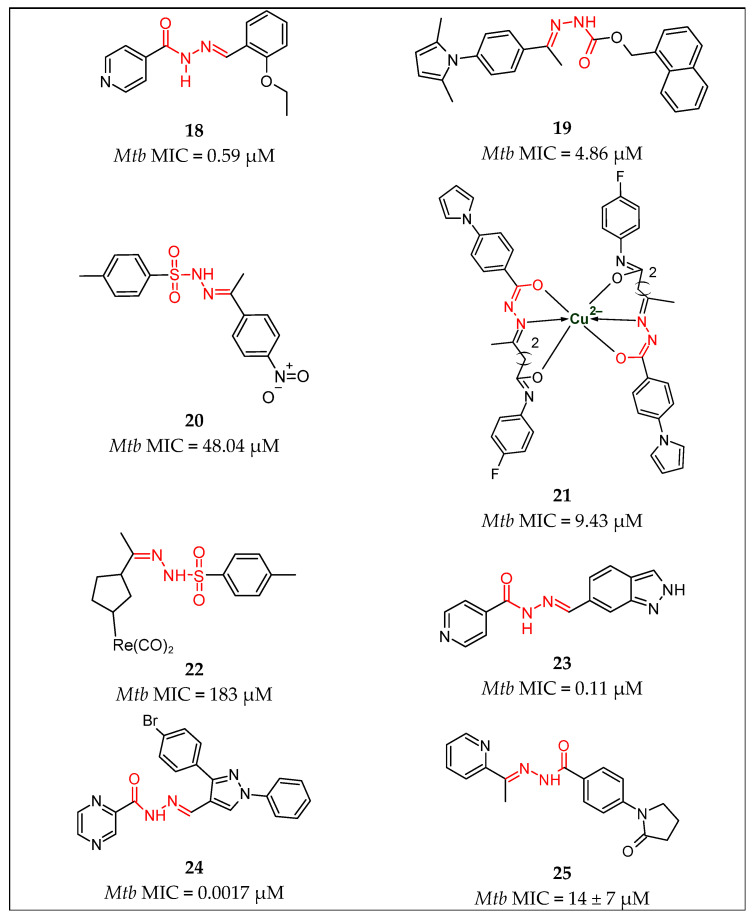
Chemical structures of compounds **18** [42], **19** [43], **20** [44], **21** [45], **22** [46], **23** [47], **24** [48] and **25** [49].

**Figure 8 pharmaceuticals-16-00484-f008:**
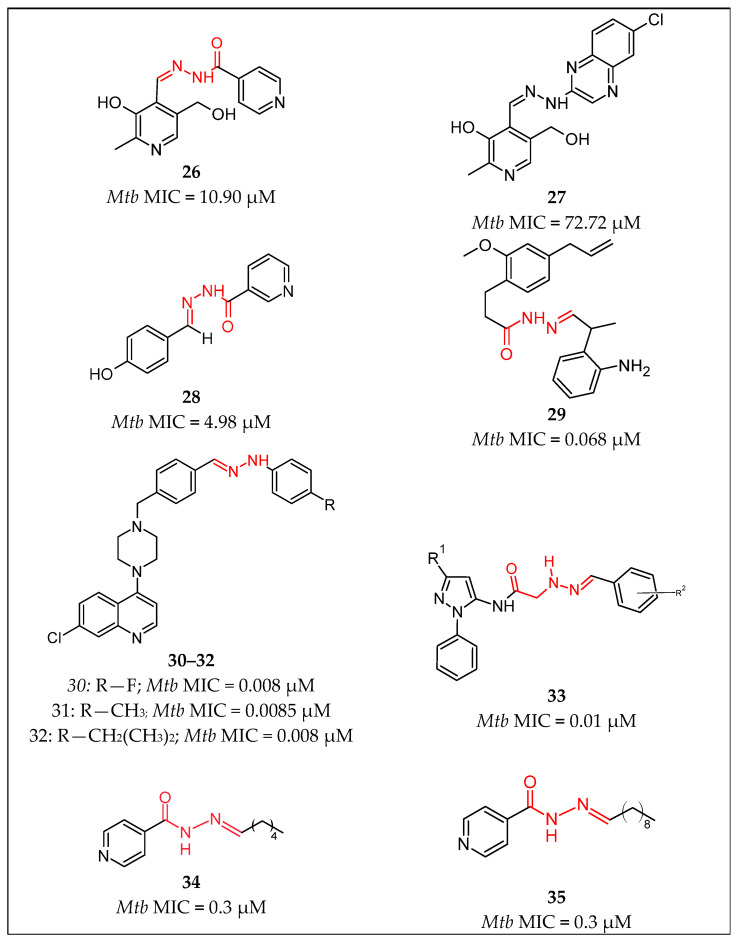
Chemical structures of compounds **26** [51], **27** [51], **28** [52], **29** [53], **30**–**32** [54], **33** [55], **34** [56] and **35** [56].

**Figure 9 pharmaceuticals-16-00484-f009:**
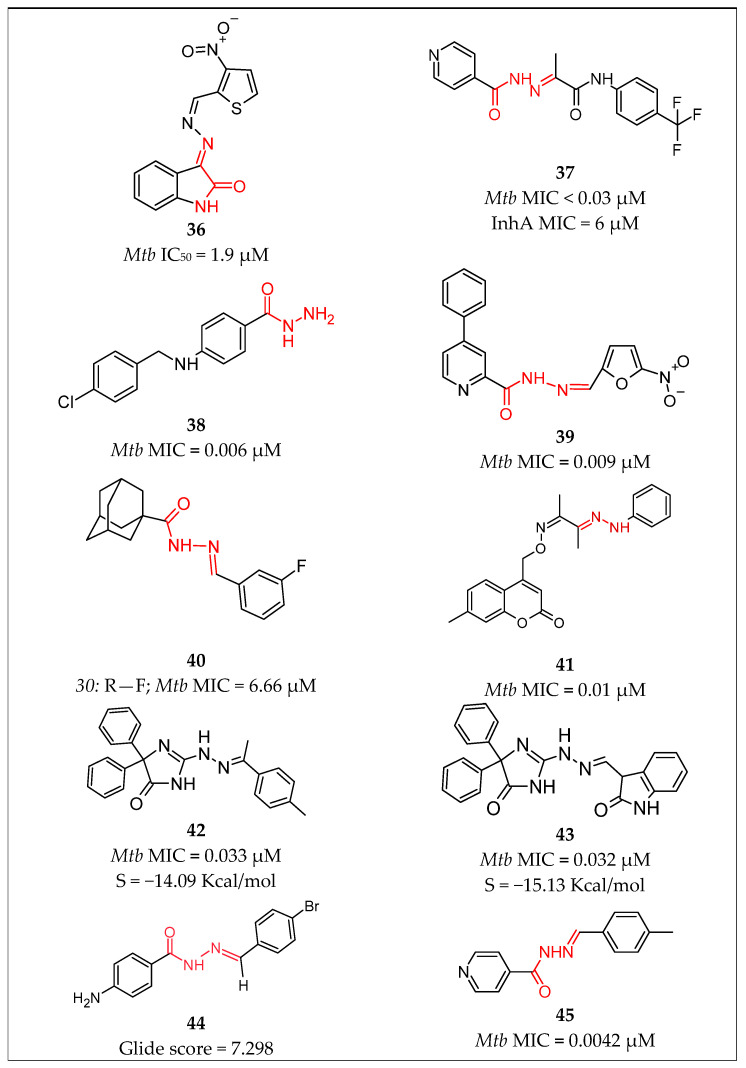
Chemical structures of compound **36** [57], **37** [58], **38** [59], **39** [60], **40** [61], **41** [62], **42** [63], 43 [63], **44** [64] and **45** [65].

**Figure 10 pharmaceuticals-16-00484-f010:**
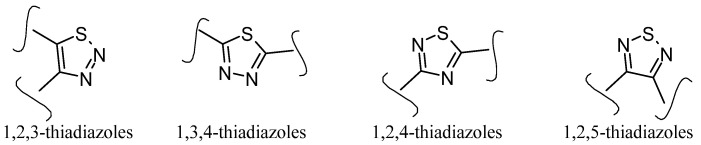
Structures of different isomeric forms of thiadiazole ring.

**Figure 11 pharmaceuticals-16-00484-f011:**
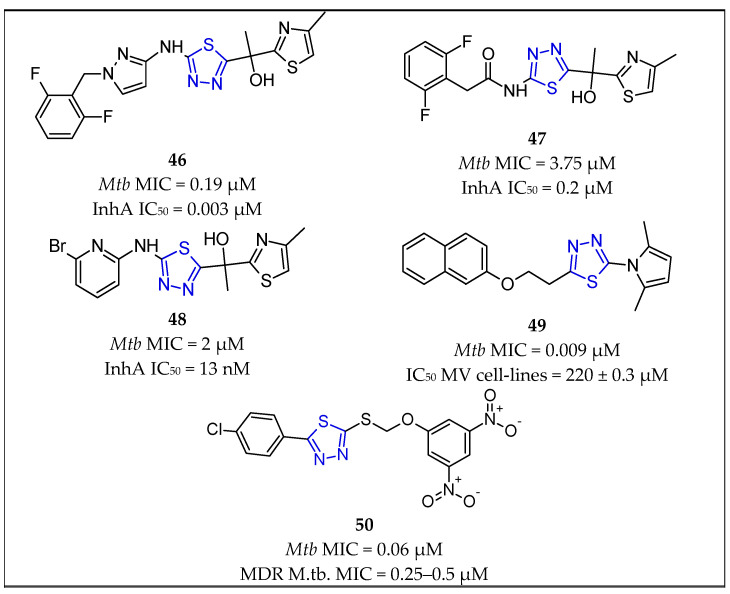
Chemical structures of thiadiazole-containing compounds **46** [68,69], **47** [8], **48** [15], **49** [70], and **50** [71].

**Figure 12 pharmaceuticals-16-00484-f012:**
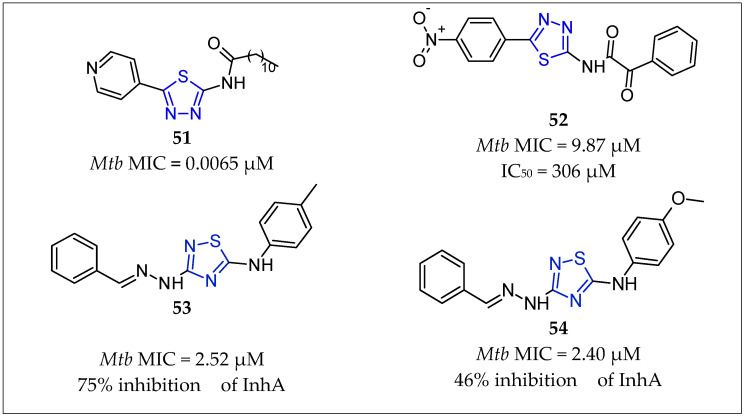
Chemical structures of thiadiazole-containing compounds **51** [72], **52** [73], **53** [12] and **54** [12].

**Table 1 pharmaceuticals-16-00484-t001:** Summary of anti-tubercular activity of hydrazone, hydrazide-hydrazone and sulfonyl–hydrazine derivatives (last 10 years).

No.		Substituents	Compd.	MIC (µM)	SI	LD_50_ (mg/kg b.w.)	InhA	References
**Hydrazide-hydrazones**	1	Pyridine/2*H*-chromene	**2**	0.17	449	-	-	Angelova et al., 2017 [28]
	2	Pyridine/(CH_2_)_15_-CH_3_	**8**	0.096	-	>5000	-	Kumar et al., 2014 [34]
	3	Pyridine/Ar	**9**	0.014	-	-	-	Pahlavani et al., 2015 [35]
	4	Pyridine/Indole	**11**	1.43	300	-	-	Velezheva et al., 2016 [37]
	5	Pyridine/Isatin	**15**	0.17	-	-	-	Santoso et al., 2021 [39]
	6	Pyridine/Ar	**16**	0.62	-	-	41% inhibition	Koçak Aslan et al., 2022 [40]
	7	Pyridine/2,1,3-benzoxadiazole	**17**	1.1	472	-	-	Fernandes et al., 2017 [41]
	8	Pyridine/Ar	**18**	0.59	-	-	-	Kumar et al., 2014 [42]
	9	Pyridine/2*H*-indazol	**23**	0.11	-	-	19% Inhibition	Oliveira et al., 2017 [47]
	10	Pyridine/Ph	**26**	10.90	-	-	-	Nogueira et al., 2018 [51]
	11	Pyridine/2-hydroxyPh	**28**	4.98	-	-	-	Sampiron et al., 2019 [52]
	12	Pyridine/(CH_2_)_4_-CH_3_	**34**	0.03	-	-		Faria et al., 2021 [56]
	13	Pyridine/(CH_2_)_8_-CH_3_	**35**	0.03	-	-	-	Faria et al., 2021 [56]
	14	Pyridine/Ph-propanamide	**37**	0.03	-	-	MIC = 6 µM	Pflégr et al., 2021 [58]
	15	Pyridine/5-nitrofuran	**39**	0.009	-	-	-	Gobis et al., 2022 [60]
	16	Pyridine/Ar	**45**	**0.004**	-	-	-	Lone et al., 2023 [65]
	17	1,2,3-Thiadiazole/Indole	**5**	0.396	1979	>2000	-	Angelova et al., 2019 [30]
	18	1,2,3-Thiadiazole/Ar	**6**	0.073	3516	>2000	-	Angelova et al., 2022 [31]
	19	Ar/2*H*-Chromene	**1**	0.13	697	-	-	Angelova et al., 2017 [28]
	20	Ar/Coumarin	**3**	0.32	>625	-	-	Angelova et al., 2017 [29]
	21	Ar/pyridin, Me	**25**	14	-	-	-	Bonnett et al., 2018 [49,50]
	22	Ar/NH-pyrazole	**33**	**0.01**	-	-	-	Padmini et al., 2021 [55]
	23	Ar/Ar	**44**	7.29	-	-	-	Senthilkumar et al., 2016 [64]
	24	R/2-(propan-2-yl)aniline	**29**	0.068	-	-	-	Rohane et al., 2020 [53]
	25	furan/Indole	**4**	0.44	>634	-	-	Angelova et al., 2019 [30]
	26	Indole/Ar	**10**	0.067	-	-	-	Cihan-Üstündağ et al., 2016 [36]
	27	Naphthalen-1-yl-methyl/pyrrole	**19**	4.86	-	-	-	More et al., 2014 [43]
	28	Adamantyl/Ar	**40**	6.66	-	-	-	Briffotaux et al., 2022 [61]
	29	4-Chlorobenzyl)amino] Benzohydrazide	**38**	**0.006**	-	-	-	Desale et al., 2022 [59]
	30	6-Chloroquinoxalin-2-yl/Ar	**27**	72.72	-	-	-	Nogueira et al., 2018 [51]
**Hydrazones**	31	Ar/Ar	**30–32**	**0.008**	-	-		Sruthi et al., 2020 [54]
	32	thiophen-2-yl/1,3-dihydro-2*H*-indol-2-one	**36**	1.9	-	-		Karunanidhi et al., 2021 [57]
	33	Ph/coumarin	**41**	0.021	-	-		Akki et al., 2022 [62]
	34	Imidazol-4-one/Ar	**42**	0.033	-	-		Abdelhamid et al., 2022 [63]
	35	Imidazol-4-one/2*H*-indol-2-one imidazol-4-one	**43**	0.032	-	-		Abdelhamid et al., 2022 [63]
**Sulfonyl hydrazones**	36	Ar/ethenylbenzene	**7**	0.0716	3380			Angelova et al., 2022 [31]
	37	*para*-tolyl/Ph *para*-tolyl/2-OMePh *para*-tolyl/2-naphtyl	**12** **13** **14**	1.25 1.25 1.25	-	-	-	Ghiano et al., 2020 [38] Ghiano et al., 2020 [38] Ghiano et al., 2020 [38]
	38	*para*-tolyl/4-nitroPh	**20**	48.04	-	-	-	Ghiya and Joshi et al., 2016 [44]
	*39*	*para*-tolyl/R_1_R_2_	**22**	183	-	-	-	Concha et al., 2017 [46]
	40	Pyrazine/1*H*-pyrazole	**24**	**0.0017**	1085.7	846.9	-	Hassan et al., 2020 [48]
**Thiadiazoles**	41	5-amino-1,3,4-thiadiazole	**46**	0.19	-	-	MIC = 0.003 μM	Ballell et al., 2010 [68] Castro et al., 2012 [69] Shirude et al., 2013 [8]
	42	5-amino-1,3,4-thiadiazole	**47**	3.75	-	-	MIC = 0.2 μM	Shirude et al., 2013 [8]
	43	5-amino-1,3,4-thiadiazole	**48**	2.00	-	-	MIC = 13 nM	Šink et al., 2018 [15]
	44	5-pyrrolyl-1,3,4-thiadiazole	**49**	**0.009**	-	-	-	of Joshi et al., 2018 [70]
	45	5-sulfanyl-1,3,4-thiadiazoe	**50**	0.06	-	-	-	Karabanovich et al., 2016 [71]
	46	5-amino-1,3,4-thiadiazol	**51**	2.25	-	-	-	Mali et al., 2020 [72]
	47	5-amino-1,3,4-thiadiazol	**52**	9.87	-	-	-	Patel et al., 2019 [73]
	48	3-benzylidenehydrazinyl-1,2,4-thiadiazol-5-amine	**53**	2.52	-	-	75% inhibition	Doğan et al., 2020 [12]
	49	3-benzylidenehydrazinyl-1,2,4-thiadiazol-5-amine	**54**	2.40	-	-	46% inhibition	Doğan et al., 2020 [12]

## Data Availability

Publicly available datasets were analyzed in this study.

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
