# Peer review of "Recent Advances in Anti-Tuberculosis Drug Discovery Based on Hydrazide–Hydrazone and Thiadiazole Derivatives Targeting InhA"

_pharmaceuticals, 2023, doi:10.3390/ph16040484_

Round 1
Reviewer 1 Report
The purpose of this review is to assess recently described hydrazide-hydrazone and thiadiazole-containing derivatives that inhibit InhA activity and have antimycobacterial effects. In addition, the mechanisms of action of currently available anti-tuberculosis drugs, including newly approved agents and molecules in clinical trials, are briefly reviewed.
Overall, it is a well-written article, but few sections must be revised prior to acceptance.
The authors have merely compiled data from the literature, and no effort has been made to compare compounds. Even with careful inspection, the activity values of different compounds are not in the same unit. How can a comparison be made during such a presentation? To conduct an accurate comparison of activities, authors should convert mg, micro gram, and mili molar values to the unit micro mol.
There are no concluding remarks regarding the direction of researchers' future work. After comparing SARs, authors should indicate which is the best chemical foundation for future research.
On the basis of all reported articles, a review article should not merely be a compilation of data, but also include the author's clear and direct opinion.
Author Response
Reviewer 1:
- The authors have merely compiled data from the literature, and no effort has been made to compare compounds. Even with careful inspection, the activity values of different compounds are not in the same unit. How can a comparison be made during such a presentation? To conduct an accurate comparison of activities, authors should convert mg, micro gram, and mili molar values to the unit micro mol.
Answer: The MIC values of all compounds have been presented into the same unit - µM. Additionally, a table with results from comparison between the structures and their activity has been introduced.
- There are no concluding remarks regarding the direction of researchers' future work. After comparing SARs, authors should indicate which is the best chemical foundation for future research.
Answer: At the end of the manuscript, we have included concluding remarks with regards to the antitubercular activity of the compounds and the further direction of our research work.
- On the basis of all reported articles, a review article should not merely be a compilation of data, but also include the author's clear and direct opinion.
Answer: It's done

Reviewer 2 Report
In this manuscript entitled: “Recent advances in anti-tuberculosis drug discovery based of hydrazide-hydrazone and thiadiazole derivatives targeting InhA”, the authors: Yoanna Teneva 1, Rumyana Simeonova *, Violeta Valcheva, and Violina Angelova * present the review focused on some hydrazide-hydrazone and thiadiazole-containing derivatives with the anti-tuberculosis effect, that inhibit InhA activity. In addition, a brief review of the mechanisms of action of currently available anti-tuberculosis drugs, including agents and molecules recently approved in clinical trials, was provided.
I believe that the paper may be suitable for publication in the MDPI journal Pharmaceuticals after addressing the following considerations listed below.
- I suggest that in the title “based of” to be replaced with “based on”:
“Recent advances in anti-tuberculosis drug discovery based on hydrazide-hydrazone and thiadiazole derivatives targeting InhA”.
- It is necessary that section “1. Introduction” be improved. I suggest the introduction of a figure with some representative anti-tuberculosis drugs/agents containing hydrazide-hydrazone and thiadiazole scaffolds. I also suggest mentioning some representative previous reviews on this topic.
- In Figure 1, in the structure of isonicotinoyl radical it is necessary that the radical (·) to be at the C atom (and not next to the C=O bond).
- Figures should be placed in the main text near the first time they are cited. For example, Figure 5 is cited in the main text in line 295 and its title in line 296. I suggest it be placed in line 242 and the title of Figure 5 in line 243. Also, at the end of the title of each figure, there needs to be a period.
- I consider that the part of the review where generalities about tuberculosis and mechanisms of action are presented is extended compared to section 3. I suggest to include in section 3 other hydrazide-hydrazone and thiadiazole containing derivatives that inhibit InhA activity and corresponding references.
Author Response
Reviewer 2:
Comment 1: I believe that the paper may be suitable for publication in the MDPI journal Pharmaceuticals after addressing the following considerations listed below.
- I suggest that in the title “based of” to be replaced with “based on”:
“Recent advances in anti-tuberculosis drug discovery based on hydrazide-hydrazone and thiadiazole derivatives targeting InhA”.
Answer: It is done!
Comment 2: It is necessary that section “1. Introduction” be improved. I suggest the introduction of a figure with some representative anti-tuberculosis drugs/agents containing hydrazide-hydrazone and thiadiazole scaffolds. I also suggest mentioning some representative previous reviews on this topic.
Answer: The Introduction is improved! In section “1. Introduction” a figure has been introduced (Figure 1), which presents the chemical scaffolds which have been reported in the literature to possess InhA inhibition activity. In addition, examples of three similar review publications have been cited below the figure.
Some representative anti-tuberculosis drugs/agents containing hydrazide-hydrazone and thiadiazole scaffolds are added. The bibliography was updated!
Comment 3: In Figure 1, in the structure of isonicotinoyl radical it is necessary that the radical (·) to be at the C atom (and not next to the C=O bond).
Answer: In Figure 1 the radical (·) is placed at the C atom even though it looks like at the C=O bond, but this is done by the used software (ACD/ChemScetch).
Comment 4: Figures should be placed in the main text near the first time they are cited. For example, Figure 5 is cited in the main text in line 295 and its title in line 296. I suggest it be placed in line 242 and the title of Figure 5 in line 243. Also, at the end of the title of each figure, there needs to be a period.
Answer: Figures 5,6,10 and 11 are moved closer to the line, in which they are first mentioned. Also, at the end of titles of all figures, a period is introduced.
Comment 5: I consider that the part of the review where generalities about tuberculosis and mechanisms of action are presented is extended compared to section 3. I suggest to include in section 3 other hydrazide-hydrazone and thiadiazole containing derivatives that inhibit InhA activity and corresponding references.
Answer: Section “2.6 New candidates in clinical trials and recently approved drugs” and Figure 3 have been removed from the text as we believe they do not have significant contribution to the topic of the manuscript.
As per your recommendation, new references, compounds and tables have been introduced in section “3.4 Hydrazide-hydrazones” to provide a more comprehensive review on the hydrazide-hydrazones activity.
The citations have been re-introduced in the text from Endnote library to keep the sequence numbers and hyperlinks to the bibliography as they were copied as plain text.
